# Evaluation and Transcriptome Analysis of the Novel Oleaginous Microalga *Lobosphaera bisecta* (Trebouxiophyceae, Chlorophyta) for Arachidonic Acid Production

**DOI:** 10.3390/md18050229

**Published:** 2020-04-26

**Authors:** Baoyan Gao, Luodong Huang, Xueqing Lei, Ge Meng, Jianguo Liu, Chengwu Zhang

**Affiliations:** 1Department of Ecology, Research Center for Hydrobiology, Jinan University, Guangzhou 510632, China; gaobaoyan1211@126.com (B.G.); hld8124@sina.com (L.H.); mydream_hxy@163.com (X.L.); 13423612446@163.com (G.M.); 2Key Laboratory of Experimental Marine Biology, Center for Ocean Mega-Science, Institute of Oceanology, Chinese Academy of Sciences, Qingdao 266071, China; jgliu@qdio.ac.cn; 3Laboratory for Marine Biology and Biotechnology, Qingdao National Laboratory for Marine Science and Technology, Aoshanwei Town, Jimo, Qingdao 266200, China

**Keywords:** *Lobosphaera bisecta*, light intensity, medium replacement, lipid, arachidonic acid, transcriptome analysis

## Abstract

Arachidonic acid (AA) is an omega-6 long-chain polyunsaturated fatty acid and is important for human health. The coccoid green microalga *Lobosphaera bisecta* has been reported to be able to accumulate high AA content under certain conditions. Nutrient management and light intensity had significant effects on the biomass and accumulation of lipids and AA in *L. bisecta* SAG2043. Both a high nitrogen concentration (18 mM) and high light intensity (bilateral light-300 μmol m^−2^ s^−1^) were beneficial to the growth of *L. bisecta*, and the replacement of culture medium further enhanced the biomass, which reached 8.9 g L^−1^. Low nitrogen concentration (3.6 mM) and high light significantly promoted the accumulation of lipids and AA. The highest lipid and AA content reached 54.0% and 10.8% of dry weight, respectively. Lipid compositions analysis showed that 88.2% of AA was distributed within the neutral lipids. We then reconstructed the lipid metabolic pathways of *L. bisecta* for the first time, and demonstrated that the upregulation of a key desaturase and elongase in the Δ6 pathway was conducive to the accumulation of fatty acids toward AA synthesis. *L. bisecta* SAG2043 exhibits high biomass, lipid and AA production. It may be a potential candidate for AA production.

## 1. Introduction

Arachidonic acid (AA) is an omega-6 (ω-6), 20 carbon-long chain polyunsaturated fatty acid (LC-PUFA) with four double bonds (5, 8, 11, 14-eicosatetraenoic acid, C20:4). It is an integral constituent of biological cell membranes and one of the major polyunsaturated fatty acids (PUFAs) that constitute the brain membrane phospholipid [1]. It is also a precursor of the synthesis of eicosanoid hormones, such as prostaglandin, thromboxane, and leukotriene [2]. In addition, AA plays an important role in immune-suppressant, and induce inflammatory responses, blood clotting and cell signaling [1]. AA can come from exogenous sources, and can also be synthesized from the essential fatty acid, linoleic acid (LA; 18:2n-6), which is provided to humans and mammals [3]. While, LA is converted to form γ-linolenic acid (GLA, C18:3) by the enzyme Δ6-desaturase which is slow. The enzyme is present only in low levels in humans [4]. Thus, it is better to supply AA to humans, not LA. The demand for abundant, safe, and edible AA as a food ingredient is imperative. The primary sources of AA are porcine liver, adrenal glands, and fish oil, etc., but the content in these sources is rather low (about 0.2%–0.5%), and as such, alternative sources are being sought [5].

Microalgae can synthesize various components, including lipids, PUFAs, pigments, and other bioactive metabolites, which can be used in health supplements, pharmaceuticals, cosmeceuticals, and biofuels [6]. Many microalgae possess a certain amount of PUFAs, which are generally deposited in the polar lipids of cell membrane systems. The contents of PUFAs are very limited and strictly regulated [6]. Although oleaginous microalgae can accumulate a high content of neutral lipids, particularly triacylglycerols (TAGs), under stressful conditions such as high light and nitrogen deficiency, the fatty acid composition of TAGs is mostly saturated and monounsaturated fatty acids [7]. LC-PUFAs of the ω-3 family, such as eicosapentaenoic acid (EPA) and docosahexaenoic acid (DHA), are rich in some microalgae, while ω-6 LC-PUFAs are relatively rare [4,7]. Microalgae that contain a relatively high content of AA are concentrated in several strains of Rhodophyta and Euglenoid, such as *Pseudochantransia* sp. SAG 19.96, *Porphyridium cruentum* SAG 1380-1e, *Rhabdomonas incurva* SAG 1271-8, and *Khawkinea quartana* SAG 1204-9, of which the AA content was reported to be 68.3%, 44.5%, 41.3%, and 34.3% of the total fatty acid, respectively [8]. It is worth noting that AA accumulation varied among different strains of the same species. Under the same culture condition, AA content was 44.5% in *Porphyridium purpureum* SAG 1380-1e, but only 3.8% in SAG 1380-1d [8]. In order to promote microalgal usage for the commercial production of PUFAs and LC-PUFA-rich oils, it is necessary to screen and select oleaginous microalgal species with the capacity to store LC-PUFA in TAGs, which would allow for more cost-effective production of PUFAs and oil. Therefore, it is urgent to find the species that can accumulate high AA levels in neutral lipids. 

*Lobosphaera incisa* (syn. *Parietochloris incisa, Myrmecia incisa*), which belongs to Trebouxiphyceae, Chlorophyta, was first considered to be the richest plant source of AA. *L. incisa* was an oleaginous microalga that stored lipids rich in AA, and the AA content could reach 60% of the total fatty acid content [6]. *Lobosphaera bisecta* (formerly known as *Myrmecia bisecta*), which also belongs to Trebouxiphyceae and is a close relative of *L. incisa*, has been shown to be able to accumulate high AA content, reaching more than 50% of the total fatty acid content under certain conditions [8,9]. However, there have been few studies on this microalga, and the information regarding lipid accumulation and fatty acid metabolism in this strain is limited. Therefore, in this study we aimed to comprehensively evaluate the effects of two different nitrogen concentrations, three different light intensities, and four different treatments of medium replacement on the biomass, lipid and AA accumulation in the *L. bisecta*. Furthermore, we performed transcriptome analysis of *L. bisecta* in order to elucidate AA and TAG metabolism under nitrogen deficiency.

## 2. Results and Discussion

### 2.1. Evaluation of Oleaginous Microalga L. bisecta SAG2043 for Lipid and AA Production

Environmental parameters, including light, temperature, and nutrient levels, strongly influence the growth and intracellular biochemical composition of microalgae [10]. Herein, we investigated the effect of two initial nitrogen concentrations in the culture medium, 18 mM and 3.6 mM, represented as high and low nitrogen concentration, respectively, three light intensities (ul-100, ul-300, bl-300; described in Section 3.2.1), and four medium replacement treatments (18 mM to 18 mM, 18 mM to 0 mM, 3.6 mM to 3.6 mM, 3.6 mM to 0 mM; described in Section 3.2.2) on the growth, lipid content, and fatty acids profiles of *L. bisecta* SAG2043.

#### 2.1.1. Effects of Different Nitrogen Concentrations, Light Intensity, and Medium Replacement on the Growth of *L. bisecta* SAG2043

Growth curves under different initial nitrogen concentrations and light intensities showed that the biomass of *L. bisecta* SAG2043 increased gradually with the culture time, but there were significant differences in the growth rate and total biomass concentration of different experimental groups (Figure 1a). Under the condition of ul-100, the difference of biomass with two different nitrogen concentrations was relatively small, and the maximum biomass of high nitrogen and low nitrogen concentration was 3.8 g L^−1^ and 3.5 g L^−1^, respectively (one-way ANOVA, *p* = 0.03 < 0.05). Under the condition of ul-300, the growth rate was higher than that under the condition of low light intensity, and the maximum biomass with high and low nitrogen concentrations was 5.0 g L^−1^ and 4.5 g L^−1^, respectively (one-way ANOVA, *p* = 0.08). To further improve the light intensity, cool white fluorescent lamps were supplied on either side of the photobioreactor, which was bl-300. Under the condition of bl-300, the growth rate was much higher compared to the culture of unilateral illumination condition (ul-100, ul-300), and the growth rate with high and low nitrogen concentrations was slightly different. The maximum biomass was eventually 5.6 g L^−1^ (18 mM) and 5.1 g L^−1^ (3.6 mM) (one-way ANOVA, *p* = 0.04 < 0.05). Microalgae are, generally, photosynthetic organisms that need light as an energy source in order to convert carbon into biomass and split water into oxygen and proton. Therefore, light intensity is one of the major limiting factors in microalgae growth, and it directly influences the biochemical composition and biomass of microalgae [5]. In our study, high light intensity promoted the growth of *L. bisecta* SAG2043 despite the culture being under high nitrogen or low nitrogen concentration. Actually, too high or too low light intensities were not beneficial to the growth of microalgae. Optimal light requirement for biomass accumulation varied between different microalgae [5]. For *Euglena gracilis*, the light intensity of 100 μmol m^−2^ s^−1^ was optimal, and the growth rate decreased over 150 μmol m^−2^ s^−1^ [11]. For *L. bisecta* SAG2043, bl-300 still had a positive effect on growth. Nitrogen is an essential element and component of nitrogen-containing compounds in microalgal cells, and plays an important role in growth, photosynthesis, and energy metabolism [12]. Nitrogen limitation in the culture can reduce the growth of microalgae. A high nitrogen concentration was beneficial to the biomass accumulation of *L. bisecta* SAG2043, especially under high light intensity.

Under the cultivation of *L. bisecta* SAG2043, the growth rate was decreased after 9 days. In order to sustain long-term cell growth, we replaced the supernatant of the culture with fresh medium at this time point (9 days) (Figure 1b). After the four different treatments of medium replacement under bl-300, the maximum biomass was eventually reached, which was 8.9 g L^−1^ (18 mM to 18 mM)^a^, 8.3 g L^−1^ (18 mM to 0 mM)^b^, 8.6 g L^−1^ (3.6 mM to 3.6 mM)^a^ and 7.6 g L^−1^ (3.6 mM to 0 mM)^c^ (one-way ANOVA; a, b, c, significant difference between treatments). Obviously, replacement of culture medium significantly enhanced the biomass accumulation of *L. bisecta* SAG2043. The biomass under fresh medium containing a high or low nitrogen supply (18 mM to 18 mM, 3.6 mM to 3.6 mM) was higher than that of fresh medium without a nitrogen supply (18 mM to 0 mM, 3.6 mM to 0 mM). This indicated that the decrease in the growth rate was mainly due to nutrient deficiency, especially nitrogen. This result was also observed in other studies about replacement cultivation [13,14,15]. Replacement with fresh medium not only supplied nutrients to promote the growth of *L. bisecta* SAG2043, but also removed the extracellular metabolites produced by algal cells and released into the old medium, which may inhibit the cell growth or induce cell death [14].

#### 2.1.2. Effects of Different Nitrogen Concentrations, Light Intensity and Medium Replacement on the Lipid Content of *L. bisecta* SAG2043

During the course of culture, the lipid content of *L. bisecta* SAG2043 increased gradually (Figure 2a). Under the same nitrogen concentration, the lipid content under a high light condition was higher than it was under a low light intensity. Under the same light intensity, the lipid content at a low nitrogen concentration was higher than that observed under a high nitrogen concentration. Under ul-100, the maximum lipid content accounted for 29.3% (18 mM) and 34.0% (3.6 mM) of the dry weight on the last day of the culture (one-way ANOVA, *p* < 0.01). Under the condition of ul-300, the lipid accumulation rate was faster than it was under low light, and the maximum lipid content reached 40.4% (18 mM) and 43.2% (3.6 mM) of the dry weight (one-way ANOVA, *p*<0.01). Under bl-300, the lipid content under high and low nitrogen concentrations reached the maximum on the last day of the culture, accounting for 42.1% (18 mM) and 46.5% (3.6 mM) of the dry weight, respectively (one-way ANOVA, *p* < 0.01). Light intensity and nitrogen concentration were important factors influencing the lipid content in the cells of *L. bisecta* SAG2043. In our study, high light intensity and low nitrogen concentration were conducive to lipid accumulation in *L. bisecta* SAG2043. In most algae, lipid metabolism was regulated by environmental variables [16]. Various studies have been conducted to investigate the effect of environment parameters and key nutrients on the lipid biosynthesis of microalgae, and the method of bioprocess engineering was the simplest, and most effective strategy used to trigger lipid accumulation in microalgae [17]. The lack of nutrients caused unfavorable situations, and the cells enhanced lipid accumulation as a response. A positive correlation between the lipid content and culture time was revealed during the comparison of lipid content in the exponential and stationary growth phase of *L. bisecta* SAG2043. As culture time was prolonged, nitrogen in the culture medium was consumed, and intracellular nitrogen decreased. Thus, cells in stationary phase are under more nitrogen stress than those in the exponential phase. This is consistent with what was reported in other studies [18,19]. However, several strains of *Tetraselmis* were shown to produce more lipids during exponential growth than in the stationary growth [20]. Light intensity is an important parameter that influences the lipid content of microalgae [16]. Many studies have been performed to show that high light intensity is needed in order to obtain higher lipid accumulation. Liu et al. reported that lipid content in *Scenedesmus* sp. was increased 11-fold when the light intensity was changed from 250 to 400 μmol photons m^−2^ s^−1^ [21]. Takeshita et al. [22] showed that eight strains of *Chlorella* accumulated more lipids with a high light intensity of 600 μmol photons m^−2^ s^−1^. In the culture system of this experiment, the maximum light intensity achieved was 300 μmol photons m^−2^ s^−1^, and we further increased the light intensity by putting cool white fluorescent lamps on either side of the photobioreactor, which was bl-300. This method added the ratio between light exposure surface area and volume of culture.

The change of total lipid content in the *L. bisecta* SAG2043 under four treatments of medium replacement (18 mM to 18 mM, 18 mM to 0 mM, 3.6 mM to 3.6 mM, 3.6 mM to 0 mM) and light intensity of bl-300 is shown in Figure 2b. The lipid content in the cells reached more than 30% of the dry weight on the day 9. After the replacement of medium, the lipid content in the nitrogen-replete experimental group (18 mM to 18 mM, 3.6 mM to 3.6 mM) first decreased and then increased. The lipid content in the nitrogen-free group (18 mM to 0 mM, 3.6 mM to 0 mM) increased gradually during the entire culture period. The lipid contents under the four treatments of replacement cultivation reached the maximum on the last day of culture, which were 36.7% (18 mM to 18 mM)^a^, 40.1% (18 mM to 0 mM)^b^, 45.8% (3.6 mM to 3.6 mM)^c^ and 54.0% (3.6 mM to 0 mM)^d^ of dry weight, respectively (one-way ANOVA; a, b, c, d, significant difference between treatments). The duration of nitrogen deprivation was also an important factor that influenced the lipid accumulation [17]. Under prolonged nitrogen stress, lipid synthesis continued, and lipid content was much higher in the fresh medium without nitrogen than it was obtained under a shorter culture time. Nitrogen-replete conditions decreased the lipid content in *L. bisecta* SAG2043. In contrast, it is unusual that nitrogen-replete condition resulted in lipid accumulation in *Isochrysis zhangjiangensis* [23]. This demonstrates that microalgae species exhibit different physiological behaviors under nitrogen stress. 

#### 2.1.3. Effects of Different Nitrogen Concentrations, Light Intensity, and Medium Replacement on the Fatty Acids Profile of *L. bisecta* SAG2043

In addition to the lipid content, a high quality fatty acids profile is also crucial in biodiesel production and nutraceutical applications [24]. The main fatty acids of *L. bisecta* SAG2043 were palmitic acid (C16:0), stearic acid (C18:0), oleic acid (C18:1), linoleic acid (C18:2), linolenic acid (C18:3), arachidonic acid (AA, C20:4), and eicosapentaenoic acid (EPA, C20:5). Under different light intensities (Figure 3), the change in the relative content of main fatty acids (% total fatty acids) under two nitrogen concentrations were similar. The content of C16:0, C18:0, C18:2 and C20:5 was decreased with the culture time, while the content of C18:1 and C20:4 was increased. Interestingly, when the AA content in *L. bisecta* SAG2043 reached 24%–26% of total fatty acid, the relative content of AA held steady with the culture time and nitrogen concentration. Compared to culture in a high nitrogen concentration (18 mM) and low light intensity (ul-100), the change in the individual fatty acid content in the culture grown under a low nitrogen concentration (3.6 mM) and high light intensity (bl-300) was even more pronounced. In general, nitrogen deficiency and high light increased the saturation level of algal fatty acids [17]. In *L. bisecta* SAG2043, the content of saturated fatty acids (SFA) and PUFAs was decreased, whereas monounsaturated fatty acids (MUFA), which was oleic acid (C18:1) in *L. bisecta* SAG2043, was increased with culture time. Oleic acid was the most abundant fatty acid produced during stationary growth phase. This is a common phenomenon in green algae [25]. However, this fatty acid composition in *L. bisecta* SAG2043 differed distinctly from that in the *L. incisa* K-1 and *L. incisa* SAG 2468, which contained a relative low proportion of oleic acid [26]. Breuer et al. [25] also reported that the ratio of valuable PUFAs, EPA, AA, and DHA to the total fatty acids in *Phaeodactylum tricornutum*, *Porphyridium cruentum*, and *Isochrysis galbana* decreased in nitrogen-deficient culture. Whereas, AA content in *L. bisecta* SAG 2043 was increased gradually until it reached to 26% of the total fatty acid. The change trend of the relative content of main fatty acids (% total fatty acids) for the four treatments of medium replacement and light intensity of bl-300 was similar to the condition without medium replacement (Appendix A).

The content of AA (% dry weight) in *L. bisecta* SAG2043 under different culture conditions showed an increasing trend with the prolongation of the culture time (Figure 4a,b). Under the condition of ul-100, the difference of AA content with two different nitrogen concentrations was small during the early stage of the culture period, and the difference increased with the culture time. The AA content reached the maximum on the last day of the culture, accounting for 3.5% (18 mM) and 4.4% (3.6 mM) of the dry weight, respectively (one-way ANOVA, *p* < 0.01). Under ul-300, the AA accumulation rate under a low nitrogen concentration was much faster than it was under a high nitrogen concentration. The highest AA content under high and low nitrogen concentration was 5.4% and 6.5% of the dry weight, respectively (one-way ANOVA, *p* < 0.01). Compared to the AA content under ul-100, the AA content with the same nitrogen concentration was higher under ul-300. Under the bl-300, the change in the AA content was similar with other conditions, and the maximum content of AA reached 8.2% (18 mM) and 8.6% (3.6 mM) of the dry weight, respectively (one-way ANOVA, *p* = 0.02 < 0.05). For the treatments of culture medium replacement under bl-300, the AA accumulation rate was slow during the early stage after replacement of the culture medium. The AA content first decreased and then increased in the treatment of 18 mM–18 mM. For the treatments of 18 mM–0 mM and 3.6 mM–3.6 mM, the AA accumulate rate increased slowly, but the AA content under the treatment of 3.6 mM–0 mM rapidly accumulated after the replacement of the old medium with fresh nitrogen-free medium. The AA content of all of the experimental treatments reached the maximum on the last day of the culture, accounting for 6.4% (18 mM–18 mM)^a^, 7.0% (18 mM–0 mM)^b^, 8.5% (3.6 mM–3.6 mM)^c^ and 10.8% (3.6 mM–0 mM)^d^ of dry weight, respectively (one-way ANOVA; a, b, c, d, significant difference between treatments). Light intensity and nitrogen concentration, as well as the replacement of medium, was able to change the AA content in the *L. bisecta* SAG2043. High light intensity and low nitrogen concentration accelerated AA accumulation within the cells, and the replacement of culture medium further promoted the accumulation of AA.

#### 2.1.4. The Proportion of AA within the Lipid Components of *L. bisecta* SAG2043

We noticed that the change in the relative content of AA within the fatty acid content in some culture conditions was small (Figure 3), whereas the AA content in terms of dry weight increased gradually. In order to explain such changes, the proportion of AA in the lipid components was determined. The lipid component was extracted from freeze-dried algae powder, and then the fatty acids within the lipid components were analyzed. The lipid composition was separated into neutral lipids (NLs), glycolipids (GLs) and phospholipids (PLs) by solid-phase silica gel column, and each lipid fraction was confirmed by thin-layer chromatography (TLC) (Appendix A). The NLs content increased gradually with culture time, and GLs and PLs content decreased (Figure 5a). GLs and PLs were major constituents of thylakoids, plasma membranes, and endoplasmic membrane systems, which perform a structural role. The NLs was mainly triacylglycerols, which served primarily as a storage form of carbon and energy. Under optimal growth conditions, microalgae synthesized fatty acids principally for esterification into membrane lipids, whereas cells accumulated NLs as energy sinks when it was under stress conditions [27]. On the last day of culture, NLs in *L. bisecta* SAG2043 reached 86.5% of the total lipids, and the relative content of GLs and PLs was 10.5% and 3.0% of the total lipids, respectively. The change in the AA distribution within the lipid composition showed that the AA proportion allocated in GLs and PLs decreased with the culture time, and AA proportion allocated in the NLs increased (Figure 5b). The percentage of AA distributed within lipid compositions was 88.2% (NLs), 11.5% (GLs) and 0.3% (PLs), respectively. AA was mainly distributed in the NLs during the later period of the culture. In most species, nitrogen deficiency not only promoted lipid accumulation but also induced the synthesis of specific lipid compositions and led to the redistribution of fatty acids [23]. Unlike many microalgae-contained PUFAs which are stored in polar lipids, *L. bisecta* SAG2043 could utilize NLs as a reservoir of AA, and simultaneously accumulated AA and NLs. This phenomenon was also observed in *Lobosphaera* (*Parietochloris*) *incisa* [7]. Moreover, Bigogno et al. [7] demonstrated that the capability to store long chain PUFAs in neutral lipids was a buffer capacity which allowed the microalgae to quickly adapt to the changing environment. 

#### 2.1.5. Effects of Different Nitrogen Concentrations, Light Intensity and Medium Replacement on the Lipid and AA Productivity of *L. bisecta* SAG2043

Volumetric total lipid productivity is a combination of the biomass concentration and lipid content. It is considered as a key parameter to evaluate lipid production by microalgal strains [25]. Due to the opposite change in total lipid content (or AA content) and biomass under high and low nitrogen concentration, the difference in lipid (or AA) productivity under the two nitrogen concentrations was not significant (Figure 6a). However, the effect of light intensity on lipid and AA productivity was significant. High light obviously improved total lipid and AA productivity, and the highest lipid and AA productivity was obtained in the culture exposed to bl-300. Under bl-300, the total lipid productivity with high nitrogen and low nitrogen concentration was 156.1 mg^−1^ L^−1^ day^−1^ and 157.0 mg^−1^ L^−1^ day^−1^, respectively (one-way ANOVA, *p* = 0.325), and the AA productivity was 30.2 mg^−1^ L^−1^ day^−1^ and 29.0 mg^−1^ L^−1^ day^−1^, respectively (one-way ANOVA, *p* = 0.056). The replacement of culture medium under bl-300 increased the biomass concentration (18 mM–18 mM, 18 mM–0 mM), whereas it prolonged the culture time, the total lipids and AA productivity, which was 136.0 mg^−1^ L^−1^ day^−1^ (18 mM–18 mM), 137.8 mg^−1^ L^−1^ day^−1^ (18 mM–0 mM) (one-way ANOVA, *p* < 0.01), and 23.8 mg^−1^ L^−1^ day^−1^ (18 mM–18 mM), 24.2 mg^−1^ L^−1^ day^−1^ (18 mM–0 mM) (one-way ANOVA, *p* = 0.01 < 0.05), respectively, were lower than those under culture without medium replacement. Both biomass and lipid content were significantly increased under low nitrogen concentration with medium replacement (3.6 mM–3.6 mM, 3.6 mM–0 mM). The total lipid and AA productivity were higher compared to the culture without medium replacement. The total lipid productivity was 164.7 mg^−1^ L^−1^ day^−1^ (3.6 mM–3.6 mM) and 171.7 mg^−1^ L^−1^ day^−1^ (3.6 mM–0 mM), respectively (one-way ANOVA, *p* < 0.01), and the AA productivity was 30.7 mg^−1^ L^−1^ day^−1^ (3.6 mM–3.6 mM), and 34.4 mg^−1^ L^−1^ day^−1^ (3.6 mM–0 mM), respectively (one-way ANOVA, *p* < 0.01) (Figure 6b). 

### 2.2. Transcriptome Analysis of the Novel Oleaginous Microalga L. bisecta SAG2043 Revealed Its Potential Mechanism for AA Synthesis

The reference transcriptome of *L. bisecta* SAG2043 was sequenced and the double-ended sequencing was conducted using HiSeq X Ten high-throughput sequencing platform. A total of 16.81 Gb of clean reads were obtained. The number of unigenes after sequence assembly was 43,440 with an N50 length of 2833 bp (Table 1). For functional annotation, a total of 10,655 unigenes were annotated (Table 1).

To analyze the transcriptional response of *L. bisecta* SAG2043 under nitrogen deficiency, the cells grown under a high nitrogen concentration (18 mM) and nitrogen-free conditions were collected on day 3 in culture, and RNA-seq for each sample were conducted (Appendix A).

Acetyl coenzyme A (CoA) is an important precursor of de novo fatty acid synthesis. The source of Acetyl CoA in chloroplasts is probably produced by the PDHC (pyruvate dehydrogenase complex) pathway. Pyruvate, which is the end-product of the Calvin cycle and glycolysis pathway, is directly catalyzed by the PDHC to form acetyl CoA (Figure 7) [28]. PDHC is an enzyme complex composed of three enzymes and is critical for the supply of acetyl CoA. The expression of the PDHC gene located in chloroplasts was up-regulated in *L. bisecta* SAG2043. Specifically, three PDHC genes were identified (unigene_bmk.11837, unigene_bmk.12164, and unigene_bmk.8585), and the expression levels of all three genes were up-regulated by more than 2-fold. PDHC provides adequate acetyl CoA to start the de novo synthesis of fatty acids, and the CO_2_ from decarboxylation of PDHC may also enter carbon fixation, which increases the available carbon sources for fatty acid synthesis.

Acetyl-CoA carboxylase (ACCase) is the first enzyme for the de novo synthesis of fatty acids and has been reported as a key rate-limiting enzyme, catalyzing the formation of malonyl-CoA from acetyl CoA and HCO_3_^−^ [29]. Four ACCase genes were identified in *L. bisecta* SAG2043 (Unigene_BMK.11695, Unigene_BMK.11856, Unigene_BMK.12421, Unigene_BMK.11794). ACCase was affected significantly by nitrogen starvation at the mRNA level, and the up-regulation of these genes may have a strong “traction” effect on the carbon flow into fatty acid synthesis. The malonyl group of malonyl-CoA is transferred to the acyl carrier protein (ACP) by the catalysis of malonyl CoA transacylase (MAT) to produce malonyl-ACP. As an important C2 unit donor in the whole fatty acid synthesis, the change of *MAT* expression was minimal. The fatty acid synthesis then goes through a cycle of reactions, including condensation, reduction, dehydration, and a re-reduction step catalyzed by fatty acid synthase (FAS), which mainly includes 3-ketoacyl-acp synthase (KAS), 3-ketoacyl-acp reductase (KAR), 3-ketoacyl-acp dehydrase (HAD), and 3-enyl-acp reductase (EAR) (Figure 7) [30]. In SAG2043, there were eight unigenes identified encoding FAS (Appendix A). After seven cycles, the generated C16:0-ACP may be assembled into glycerolipids by an acyltransferase in chloroplasts or elongated to C18:0-ACP by KAS [30]. The first double bond is then introduced by the fatty-acyl-ACP desaturase (FAD), which may catalyze the formation of C16:1-ACP or C18:1-ACP. In SAG2043, five unigenes encoding FAD were identified, and they all used C18:0-ACP as a substrate. FAD, which catalyzed C16:0-ACP, was not found in the transcriptome. Accordingly, C16:1 could hardly be detected in the *L. bisecta* SAG2043, which was consistent with the composition of fatty acids. 

The reaction catalyzed by the fatty-acyl-ACP thioesterase (FAT) hydrolyzes acyl-ACP to form free fatty acids, which will be output to the endoplasmic reticulum, where the reactions of carbon chain elongation, desaturation, and esterification occur [31]. There are two classes of thioesterases, FATA and FATB, which are responsible for hydrolyzing unsaturated and saturated acyl-ACPs, respectively [30]. In *L. bisecta* SAG2043, the expression of FATA with C18:1-ACP as the substrate was up-regulated significantly (Unigene_BMK.38890). Overall, the up-regulation and preference of FAD and FATA may be the reason that the oleic acid content was greatly increased and became the most abundant fatty acid under nitrogen deficiency in *L. bisecta* SAG2043. As such, it provided a precursor for subsequent long-chain PUFAs synthesis.

The fatty acid profile in *L. bisecta* SAG 2043 mainly included oleic acid (C18:1), linoleic acid (C18:2), linolenic acid (C18:3), AA (C20:4), and EPA (C20:5). Based on transcriptome annotation, we reconstructed the synthesis pathway of PUFAs in *L. bisecta* SAG 2043 (Figure 7). There were two different pathways identified for the synthesis of AA (C20:4, ω-6) [32]. The first one was the conventional Δ6-pathway and the other one was the alternative Δ8-pathway in protists and some microalgae [33]. In plants, the synthesis of PUFAs starts with the formation of fatty acids catalyzed by an FAS complex in plastids. Stearic acid (SA, C18:0) is desaturated to oleic acid (OA, C18:1^Δ9^) by Δ9-desaturase [32], and then Δ12-desaturase is converted OA to linoleic acid (LA, C18:2^Δ9,12^, ω-6). In the conventional Δ6-pathway, the LA is converted to γ-linolenic acid (GLA, C18:3^Δ6,9,12^) by Δ6-desaturase, and GLA is then elongated to produce dihomo-γ-linolenic acid (DGLA, C20:3^Δ8,11,14^) by Δ6-elongase. Finally, Δ5-desaturase performs one more desaturation to produce AA (C20:4^Δ5,8,11,14^). In the alternative Δ8-pathway, the Δ9-elongase converts LA to form eicosadienoic acid (EDA, C20:2^Δ11,14^). Subsequently, EDA is desaturated to generate DGLA by Δ8-desaturase, which then yields AA by Δ5-desaturase. However, there was no Δ8-desaturase gene identified in *L. bisecta* SAG 2043. AA synthesis in *L. bisecta* SAG 2043 was by the conventional Δ6-pathway. According to the transcriptome analysis, there were 15 unigenes annotated as desaturases in *L. bisecta* SAG2043. Among them, the expression of seven genes was up-regulated, and three genes were down-regulated under the condition of nitrogen deficiency (Appendix A, Appendix A). In *L. bisecta* SAG2043, there were no unigenes identified as a Δ17 desaturase, which illustrated that EPA may mainly form through an omega-3 pathway. Furthermore, it cut off the pathway from AA to EPA, so it was conducive to the accumulation of AA.

Nitrogen deficiency can significantly promote lipid accumulation (mainly TAG) in most microalgae. Acyl groups from Acyl-CoA are transferred to a different position of the glycerol-3-phosphate (G3P) involved in the four enzymatic steps, including glycerol-3-phosphate acyltransferase (GPAT) and lyso-phosphatidic acid acyltransferase (LPAT) to transfer the first and second acyl group into the sn-1 and sn-2 position of G3P to generate phosphatidic acid (PA), followed by the dephosphorylation of phosphatidic acid phosphatase (PAP) to produce DAG, and a third acylation by diacyglycerol acyltransferase (DGAT). This is called the Kennedy pathway, which is shown in Figure 7. The classical Kennedy pathway is crucial for TAG synthesis in many species. According to the transcriptome analysis, the genes encoding three enzymes involved in the DAG synthesis pathway were identified in *L. bisecta* SAG2043 (Appendix A). GPAT included two types—one located in the chloroplast and another located in the endoplasmic reticulum. There were six genes identified as LPATs in *L. bisecta* SAG2043. Among them, the expression of four genes was up-regulated. It was previously reported that overexpression of the LPAT gene from *Brassica napus* in *Arabidopsis* resulted in a significant increase in the TAG content in the seeds [34]. The up-regulation of LPAT may play a certain role in promoting TAG accumulation. Four PAP genes were identified in *L. bisecta* SAG2043, and all four transcripts were up-regulated. There are two main pathways for eukaryote to synthesize TAG from DAG depending on the acyl donor. One is the acyl-CoA dependent pathway, and the other is the acyl-CoA independent pathway. Both pathways use DAG as an acyl receptor. DGAT is a key enzyme that catalyzes the last and committed step in the acyl-CoA dependent pathway, which adds an acyl group from acyl-CoA to the *sn*-3 position of DAG. Two types of DGAT have been identified in microalgae. According to the transcriptome analysis of *L. bisecta* SAG2043, five unigenes encoding Type-2 DGAT (DGAT2) and two unigenes encoding Type-1 DGAT (DGAT1) were identified. The transcription abundance of DGAT2 was generally higher than that of DGAT1, and the expression level of all DGAT2 was up-regulated under nitrogen deficiency, indicating that DGAT2 plays a major role in TAG synthesis. Subcellular localization of DGAT was poorly understood. Thus, the synthesis of TAG in multiple organelles needs to be further explored. The acyl-CoA independent pathway uses phospholipids, such as phosphatidylcholine (PC), as acyl donors catalyzed by phospholipid:DAG acyltransferases (PDAT) to synthesize TAG. It has been reported that PDAT was a multifunctional enzyme in *C. reinhardtii* [35]. It has been shown to mediate membrane lipid turnover and TAG synthesis. Under low nitrogen stress, membrane lipids of algae degraded gradually with the accumulation of TAG, and the fluidity became worse [35]. Two PDAT genes were predicted in *L. bisecta* SAG2043, and the expression of PDAT with high transcriptional abundance was up-regulated by more than two times (group1_unigene_bmk.12161), which indicated that the PDAT pathway also contributed to the accumulation of TAG in *L. bisecta* SAG2043.

## 3. Materials and Methods

### 3.1. Algal Strain and Culture Conditions

*Lobosphaera bisecta* SAG2043 was obtained from Culture Collection of Algae at Göttingen University, Göttingen, Germany, and deposited in our laboratory at Jinan University, Guangzhou, China. Cultures were maintained in modified BG-11 medium [36]. Initial synchronized cultures were grown in 250 mL Erlenmeyer flask with 150 mL working volume. The temperature was maintained at (24 ± 1) ℃.

### 3.2. Experimental Design

#### 3.2.1. Interactive Effects of Nitrogen Concentrations and Light Intensities on the Growth of *L. bisecta* SAG2043

The strain *L. bisecta* SAG2043 was cultured in mBG-11 medium, with sodium nitrate used as the nitrogen source, at two nitrogen concentrations of 18 mM and 3.6 mM. Three lighting conditions were conducted as following: unilateral low light illumination of 100 μmol m^−2^ s^−1^ (ul-100), unilateral high light illumination of 300 μmol m^−2^ s^−1^ (ul-300), and bilateral high light illumination of 300 μmol m^−2^ s^−1^ (bl-300). The algae seed in logarithmic growth phase was collected by low-speed centrifugation at 1000× *g* for 3 min and inoculated at an initial concentration of OD_750_ = 0.7 ± 0.01 in a Ø6.0 cm × 60 cm bubble column glass photobioreactors (1.3 L working volume). The cultures were aerated with 1% CO_2_-enriched compressed air and exposed to 24 h of continuous illumination. The temperature was maintained at 24 ± 1 ℃. Each experimental condition was performed in triplicate. The whole culture was grown for 15 days.

#### 3.2.2. The Effects of Different Nitrogen Concentrations and Medium Replacement Treatments on the Growth of *L. bisecta* SAG2043

The mBG-11 medium was set with two initial nitrogen concentrations, 18 mM and 3.6 mM. The light condition was ul-300. After 9 days of cultivation, the whole cultures were collected and centrifugated for removal of the old culture medium. The collected algal paste was reinoculated into the same column photobioreactors with fresh medium and cultivated for another 15 days. There were four medium replacement treatments. (1) The initial medium with a nitrogen concentration of 18 mM was replaced with fresh medium which also had a nitrogen concentration of 18 mM (18 mM–18 mM); (2) The initial medium with a nitrogen concentration of 18 mM was replaced with nitrogen-free fresh medium (18 mM–0 mM); (3) The initial medium with a nitrogen concentration of 3.6 mM was replaced with fresh medium containing 3.6 mM of nitrogen (3.6 mM–3.6 mM). 4) The initial medium with a nitrogen concentration of 3.6 mM was replaced with nitrogen-free fresh medium (3.6 mM–0 mM).

### 3.3. Analysis Methods

#### 3.3.1. Biomass Measurement

A 10 mL culture sample was filtered through pre-weighed glass fiber filter membrane with 0.45-μm of pore size (dry weight, W1) by vacuum filter. The filter membranes containing algal cells were then dried in an oven at 105 °C to constant weight (dry weight, W2). The dry weight of the algal cells was then calculated based on the difference between W2 and W1, divided by the volume of sampled algal suspension.

#### 3.3.2. Lipid Extraction and Determination

Total lipid extraction was conducted based on method described by Khozin-Goldberg [37] with some modifications. About 50–80 mg of freeze-dried algal powder was extracted with 2 mL of dimethyl sulfoxide -methanol mixture (V:V = 1:9) in a 50 °C water bath for 1.5 h. The mixture was then centrifuged, and the supernatant was collected. The residue was re-extracted with 4 mL diethyl ether-hexane mixed solution (V:V = 1:1) in an ice bath for 1.5 h and then centrifuged. The supernatant was collected into the same glass vial and the extraction process was repeated. The determination of lipid content was by gravimetric method and the detail was described in Gao et al. [36].

#### 3.3.3. Fatty Acids Analysis

A total of 25 mg dry weight of freeze-dried biomass was added to 2 mL methanol solution comprising 2% H_2_SO_4_ (V/V) in a small vial, and 0.25 mg heptadecanoic acid (Sigma Chemical Co., St. Louis, MO, USA) was added which was used as an internal standard. Then the vial was filled with argon gas. The mixture was incubated in a water bath at 80 °C for 1.5 h in order to promote the transmethylation of fatty acids. The reaction was then quenched by adding the mixture of 1 mL H_2_O and 1 mL hexane. The solution was centrifuged at 3500 rpm (2740× g) for 5 min, and the upper layer was then dried with N_2_, followed by addition of 100 μL hexane. Heptadecanoic acid (Sigma Chemical Co., St. Louis, MO, USA) was used as an internal standard. The fatty acid methyl esters (FAMEs) were analyzed with gas chromatography-flame ionization detector (GC-FID) on an Agilent Gas Chromatograph (Agilent 6890N GC, Agilent Technologies, Palo Alto, CA, USA) and authentic standards. Detailed procedure for GC analysis has been described by Gao et al. [36].

#### 3.3.4. Separation of Lipid Fractions and Quantification

The aforementioned lipid extracts were loaded on a solid-phase silica gel column (Sep-Pak Plus Silica, Waters) and then separated to neutral lipids (NLs), glycolipids (GLs), and phospholipids (PLs) using the method described by Christie and Han [38]. The column was equilibrated with chloroform, and then, the lipid extracts were dissolved in chloroform and loaded onto the column. Three sequential eluting solvents were used that was chloroform isolated the neutral lipids (NLs), acetone isolated the glycolipids (GLs), and methanol isolated phospholipids (PLs), respectively. Each fraction was dried under N_2_ flow and weighed. Each lipid fraction was confirmed by thin-layer chromatography (TLC) on silica gel. The developing solvent for NLs was hexane: diethyl ether: acetic acid (80:20:1) [39]. The mobile phase for GLs and PLs was chloroform: methanol: H_2_O (25:10:1) [39].

#### 3.3.5. RNA Extraction, Library Construction, Sequencing, Assembly and Functional Annotation

To learn the transcriptome changes of *L. bisecta* SAG2043 during the lipid accumulation, RNA-seq analysis was performed using +N (18 mM, NR) and −N (0 mM, NF) samples derived from day 3 cultures. The total RNA of samples was extracted using RNAiso Plus (TaKaRa Biotech Co., Beijing, China). The detailed procedure of cDNA library construction had been described in Huang et al. [40]. RNA sequencing of each sample was conducted using an Illumina HiSeq 4000 by BioMarker Technologies Co. (Beijing, China). The raw files were available from the NCBI SRA database under the accession number: PRJNA594119. The transcriptome was assembled using Trinity software [41]. The assembled genes were annotated using the BLASTx with an E-value threshold of 1.0 × 10^−5^ against the databases as follows: NR (NCBI non-redundant protein sequences), COG (Clusters of Orthologous Groups of proteins), Swiss-Prot, KEGG (Kyoto Encyclopedia of Genes and Genomes), and GO (Gene Ontology).

#### 3.3.6. Quantitative Real-Time Polymerase Chain Reaction (RT-qPCR) Analysis

To validate the expression levels of some of the genes, RT-qPCR analysis was performed using +N (18 mM, N) and −N (0 mM, NF) samples derived from day 0, day 3 and day 6 cultures. The qPCR primers (Appendix A) were designed using Primer Premier 6.0 software. The RT-qPCR was performed on CFX96 Touch (Bio-Rad, Hercules, CA, USA) with PrimeScript™ RT reagent kit and TB Green™ Premix Ex Taq™ II (TaKaRa Biotech Co., Beijing, China) according to the manufacturer’s protocols. Samples were performed in triplicate. RT-PCR reactions were performed using a Step One Plus Real-Time PCR. The relative mRNA levels were normalized to the level of 18S rRNA gene in each sample and expressed as values of relative expression compared to that of the day 0 group. Relative levels of target mRNAs were determined using the 2^−ΔΔCt^ method and normalization [40,42].

#### 3.3.7. Statistical Analysis

Statistics were performed using SPSS (version 16.0) statistical software (SPSS Inc., Chicago, IL, USA). Data was considered significantly different at *p* < 0.05 (one-way ANOVA, two-way ANOVA with Duncan test).

## 4. Conclusions

The coccoid green microalga *Lobosphaera bisecta* SAG 2043 has enormous potential for the production of lipids and AA. A low concentration of nitrogen and high intensity of light significantly promoted the accumulation of lipids and AA. The highest lipid and AA content reached 54.0% and 10.8% of the dry weight of *L. bisecta* SAG 2043, respectively. Neutral lipids comprised 86.5% of the total lipids, and 88.2% of AA was distributed within the neutral lipids. We reconstructed the lipid metabolic pathways of *L. bisecta* SAG 2043 for the first time. Transcriptome analysis showed that multi-level regulation ensured the conversion efficiency of carbon to the synthesis of fatty acids, and the up-regulation of key desaturases and elongases within the Δ6 pathway was conducive to the accumulation of fatty acids towards the synthesis of AA. The synthesis of oleic acid particularly ensured the supply of precursor to AA synthesis. Although *L. bisecta* SAG2043 has demonstrated a strong ability to accumulate oleic acid, its ability to convert oleic acid into AA was not as efficient as *Lobosphaera incisa*. Enhancing the ability of *L. bisecta* SAG 2043 to convert oleic acid to AA would improve the prospect of AA production from *L. bisecta* SAG2043.

## Figures and Tables

**Figure 1 marinedrugs-18-00229-f001:**
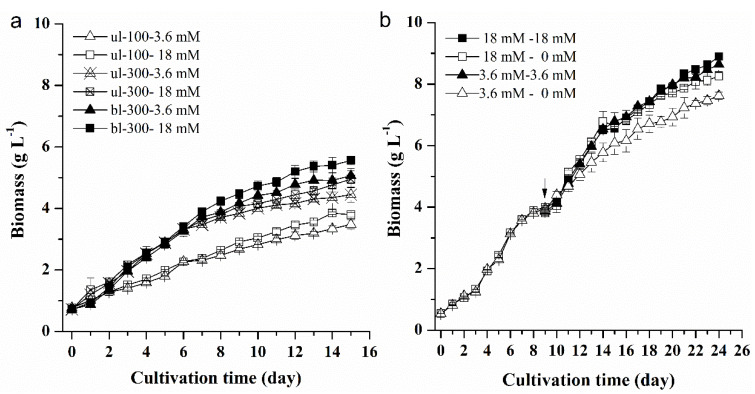
The effect of different initial nitrogen concentrations and light intensities (**a**), and different medium replacement treatments (**b**) on the growth of *L. bisecta* SAG2043; ul, unilateral illumination; bl, bilateral illumination; 100, 300, light intensity (μmol m^−2^ s^−1^); 18 mM, 3.6 mM, nitrogen concentration; 18 mM–18 mM, medium replacement from initial nitrogen concentration of 18 mM to fresh medium of 18 mM on day 9; 18 mM–0 mM, medium replacement from initial nitrogen concentration of 18 mM to fresh medium of 0 mM on day 9; 3.6 mM–3.6 mM, medium replacement from initial nitrogen concentration of 3.6 mM to fresh medium of 3.6 mM on day 9; 3.6 mM–0 mM, medium replacement from initial nitrogen concentration of 3.6 mM to fresh medium of 0 mM on day 9.

**Figure 2 marinedrugs-18-00229-f002:**
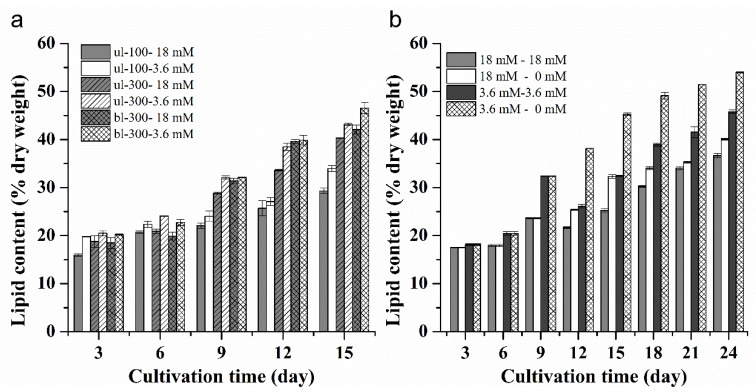
The effect of different initial nitrogen concentrations and light intensities (**a**), and different medium replacement treatments under bl-300 (**b**) on lipid content of *L. bisecta* SAG2043.

**Figure 3 marinedrugs-18-00229-f003:**
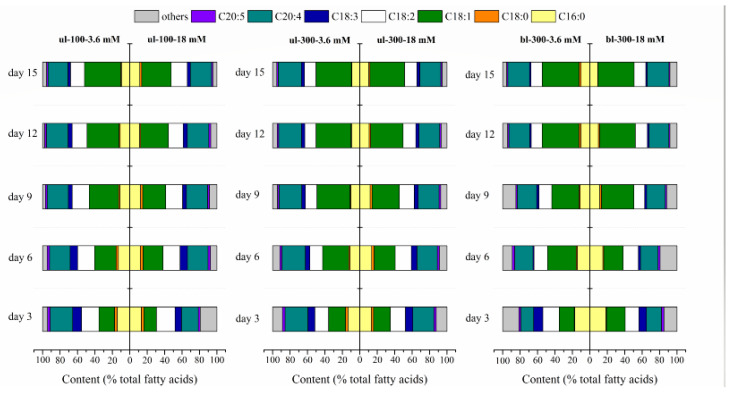
Change in the fatty acid content (% total fatty acids) of *L. bisecta* SAG2043 with two nitrogen concentrations (18 mM, 3.6 mM) under different light intensity (ul-100, ul-300, bl-300) over a time course.

**Figure 4 marinedrugs-18-00229-f004:**
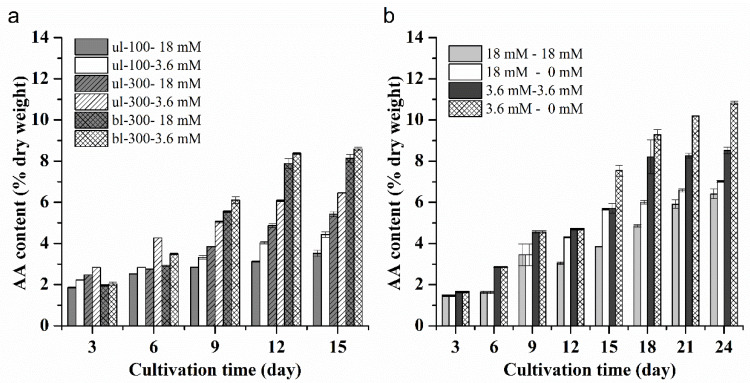
Change in the AA content of *L. bisecta* SAG2043 under different culture conditions (different initial nitrogen concentrations and light intensities (**a**), and different medium replacement treatments (**b**)) over a time course.

**Figure 5 marinedrugs-18-00229-f005:**
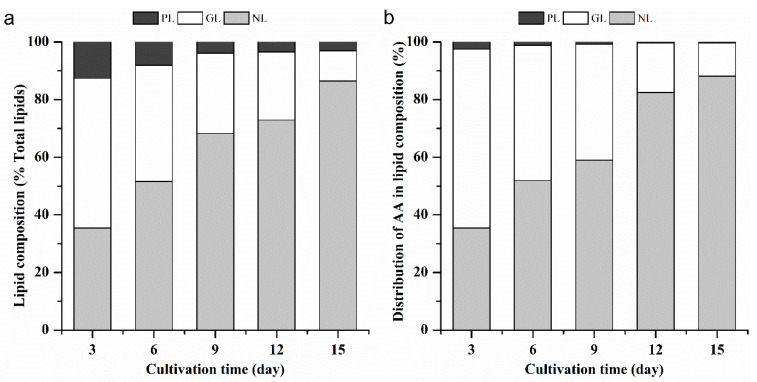
Change in the contents of each lipid composition (neutral lipid (NL), glycolipids (GL) and phospholipids (PL)) (**a**) and arachidonic acid content in different lipid fractions (**b**) under low nitrogen concentration (3.6 mM) over a time course.

**Figure 6 marinedrugs-18-00229-f006:**
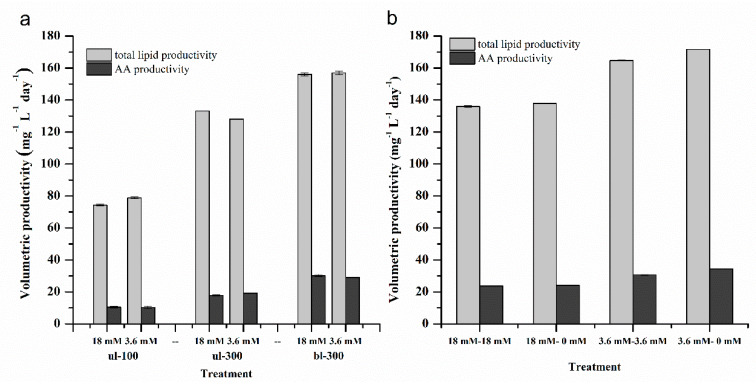
Change in the productivity of the volumetric total lipid and arachidonic acid of *L. bisecta* in different culture conditions (different initial nitrogen concentrations and light intensities (**a**), and different medium replacement treatments (**b**)) over a time course.

**Figure 7 marinedrugs-18-00229-f007:**
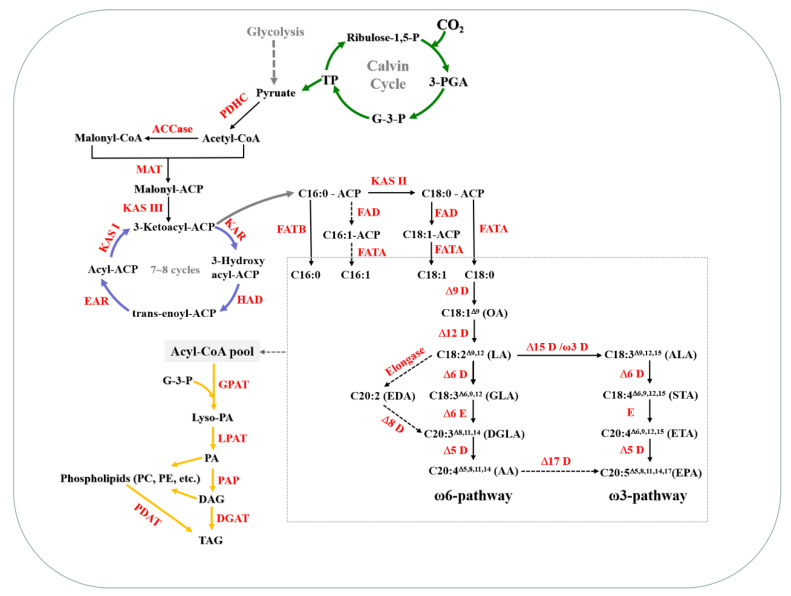
Fatty acid and triacylglycerol (TAG) biosynthesis pathways in *L. bisecta* SAG2043. Enzyme abbreviations are as follows: 3-PGA, 3-phosphoglycerate; G-3-P, glycerol-3-phosphate; PDHC, pyruvate dehydrogenase complex; ACCase, acetyl-CoA carboxylase; MAT, malonyl CoA transacylase; ACP, acyl carrier protein; KAS, 3-ketoacyl-acp synthase; KAR, 3-ketoacyl-acp reductase; HAD, 3-ketoacyl-acp dehydrase; EAR, 3-enyl-acp reductase; FAD, fatty-acyl-ACP desaturase; FAT, fatty-acyl-ACP thioesterase; D, desaturase; E, elongase; GPAT, glycerol-3-phosphate acyltransferase; LPAT, lyso-phosphatidic acid acyltransferase; PA, phosphatidic acid; PAP, phosphatidic acid phosphatase; DAG, diacylglycerol; DGAT, diacyglycerol acyltransferase; PC, phosphatidylcholine; PE, phosphatidylethanolamine; PDAT, phospholipid:DAG acyltransferases.

**Table 1 marinedrugs-18-00229-t001:** The de novo assembly and annotation of transcriptome of *L. bisecta* SAG2043.

Assembly	Results	Annotation	Results
Total Number	43,440	COG	4690
Total Length (bp)	41,805,231	GO	5695
Mean Length (bp)	962.37	KEGG	4404
N50 Length (bp)	2833	KOG	6267
200–300	17,294 (39.81%)	Pfam	8398
300–500	12,004 (27.63%)	Swiss-Prot	6080
500–1 kbp	5146 (11.85%)	eggNOG	9068
1 kbp–2 kbp	2930 (6.74%)	Nr	10,116
>2 kbp	6066 (13.96%)	All annotated	10,655

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
