# Peer review of "Evaluation and Transcriptome Analysis of the Novel Oleaginous Microalga Lobosphaera bisecta (Trebouxiophyceae, Chlorophyta) for Arachidonic Acid Production"

_marinedrugs, 2020, doi:10.3390/md18050229_

Round 1
Reviewer 1 Report
The manuscript submitted by Gao et al, described an original work which fits with the main interests of the Journal. The manuscript is well written and organized but there are several major issues that weakness the quality of the work and need to be addressed :
Major concerns:
1.Authors needs to perform statistical analysis for all data to confirm if results are significantly different
2.Clarify why AA acid is need in the diet. Considered that in the modern society the diets are imbalance of both FA, usually provide high amount of omega 6 or reduced amount of omega 3. Complete introduction section.
3. Results of the section 2.1.4. The proportion of AA within the lipid components of L. bisecta SAG204 and correspondent experimental section lack a lot of information: How did you check the composition and purity of each fraction obtained by solid phase extraction?
What was the amount of lipid extract loaded on the columns? What was the amount of eluent used to separate each fraction? what was the recovery of each fraction? How many analysis? How did you confirm the composition in neutral lipids (NLs), glycolipids (GLs) and phospholipids (PLs) and the purity of each faction ?
Also how did you measure the AA and other FA in each fraction? Did you analyze the FA in each fraction by GC-FID? Please include the data obtained in supplementary files as well as standard deviations and statistical analysis. I strongly recommend to do mass spectrometry (MS) analysis to confirm the composition in FA of the molecular species of each fraction. Only with MS data you can be sure that the AA is linked to TAGs and not to other NL as esterified sterol.
Also it was quite strange the lipid extraction protocol used (using DMSO and methanol)? Why did not use more widely used methods in lipid analysis from algae as with methanol/chloroform ( Bligh and dyer method) as reviewed by Kumar( https://doi.org/10.3389/fenrg.2014.00061) or with methanol:methyl tert-butyl ether (MTBE) ( Kokabi et al, Plant Science, 2019, 95-115) .
4.In table 1-5 : data are expressed as % of total FA ( see lines 206). But the sum of SFA and PUFA is much lower than 100%. What are the other FA?
Minor corrections:
- Lines 41-41-“ AA cannot be synthesized by humans
and mammals, and it must be provided by exogenous sources, such as food supplement “Rephrase this sentence for example as following:
“AA can come from exogenous sources but can also be synthesed from the essential FA, then 18:2 FA provided by diet.
-Line 48. Correct to “…in health….”
-Line 63-64- Why Comparing with macroalgae Gracilaria Verrucosa? Delete this sentence because micro and macroalgae have quite different biochemical composition.
-Line 121- Growth rate was decreased after 9 days of cultivation not only under Bl-300 but also under L 300.
-line 201-202- need references
- Tables 1-5 :indicate how FA are expressed in the legends. Units are missing. Include also MUFA content.
Line 181 and others and Fig 2. Indicate what was the light intensity used
Line 274” In order to explain such changes, the proportion of AA in the lipid
275 content was determined” did you mean lipid components?
Line 510- GC-FID or GC-MS?
Author Response
Response to Reviewer 1 Comments
The manuscript submitted by Gao et al, described an original work which fits with the main interests of the Journal. The manuscript is well written and organized but there are several major issues that weakness the quality of the work and need to be addressed:
Response: We would like to express our sincere thanks to you for the constructive and positive comments. Point by point responses to the comments are listed below.
Major concerns:
1. Authors needs to perform statistical analysis for all data to confirm if results are significantly different.
Response 1: Statistical analysis has been added in the revised manuscript.
2. Clarify why AA acid is need in the diet. Considered that in the modern society the diets are imbalance of both FA, usually provide high amount of omega 6 or reduced amount of omega 3. Complete introduction section.
Response 2:The introduction has been revised “AA can come from exogenous sources, and can also be synthesize from the essential fatty acid, linoleic acid (LA; 18:2n-6), which is provided to humans and mammals [3]. While, LA is converted to form γ-linolenic acid (GLA, C18:3) by the enzymeΔ6-desaturase which is slow. The enzyme is present only in low levels in humans [4]. Thus, it is better to supply AA to humans, not LA.”
3. Results of the section 2.1.4. The proportion of AA within the lipid components of L. bisecta SAG204 and correspondent experimental section lack a lot of information: How did you check the composition and purity of each fraction obtained by solid phase extraction?What was the amount of lipid extract loaded on the columns? What was the amount of eluent used to separate each fraction? what was the recovery of each fraction? How many analysis? How did you confirm the composition in neutral lipids (NLs), glycolipids (GLs) and phospholipids (PLs) and the purity of each faction ?
Also how did you measure the AA and other FA in each fraction? Did you analyze the FA in each fraction by GC-FID? Please include the data obtained in supplementary files as well as standard deviations and statistical analysis. I strongly recommend to do mass spectrometry (MS) analysis to confirm the composition in FA of the molecular species of each fraction. Only with MS data you can be sure that the AA is linked to TAGs and not to other NL as esterified sterol.
Also it was quite strange the lipid extraction protocol used (using DMSO and methanol)? Why did not use more widely used methods in lipid analysis from algae as with methanol/chloroform ( Bligh and dyer method) as reviewed by Kumar ( https://doi.org/10.3389/fenrg.2014.00061) or with methanol: methyl tert-butyl ether (MTBE) ( Kokabi et al, Plant Science, 2019, 95-115) .
Response 3: I am sorry we just do the preliminary fractionation of lipid extracts, and the method was according to “Christie, W.W.; Han, X. Analysis of simple lipid classes. In Lipid Analysis, 4rd ed.; Christie, W.W., Han, X., Eds.; Woodhead Publishing, 2012; pp. 69–90.” In the literature, the method was presented as “A short column of silica gel (about 1 g) is prepared in a glass disposable Pasteur pipette plugged with solvent-washed cotton wool. The column is conditioned by elution with chloroform (5 mL), and about 30 mg of lipid in the minimum volume of chloroform can be applied to this. Elution with chloroform or diethyl ether (10 mL) yields the simple lipids, acetone (10 mL) gives a glycolipid fraction, and methanol (10 mL) yields the phospholipids. Solvents are best allowed to flow under gravity.” We used the solid-phase silica gel column (Sep-Pak Plus Silica, Waters). The amount of lipid extract load on the column was according to the product description of the column, which was not exceed 50 mg. The amount of eluent used to separate each fraction can be according to the colour of each fraction on the column. Each fraction was dried under N2 flow, and then it was weighed to do fatty acid analysis. The fatty acid analysis of lipid fraction was same to the fatty acid analysis of algae powder. We do analyze the FA in each fraction by GC-FID, not by GC-MS. We are not sure that the AA is linked to TAGs. In our manuscript, we present that AA was distributed within the neutral lipids. The lipid extraction method was according to the literature “Khozin-Goldberg, I.; Shrestha, P.; Cohen, Z. Mobilization of arachidonyl moieties from triacylglycerols into chloroplastic lipids following recovery from nitrogen starvation of the microalga Parietochloris incisa. Biochimica et Biophysica Acta (BBA)-Molecular and Cell Biology of Lipids 2005, 1738, 63–71.” We had compared to this method with methanol/chloroform ( Bligh and dyer method). The DMSO used in this method has strong cell penetration ability which can improve the extraction efficiency. The methanol is used to extract polar lipid, and the diethyl ether-hexane mixed solution is used to extract non-polar lipid. This method has high lipid extraction efficiency and is sample.
4. In table 1-5: data are expressed as % of total FA ( see lines 206). But the sum of SFA and PUFA is much lower than 100%. What are the other FA?
Response 4: For the FAME analysis by GC-FID, each fatty acid was identified by comparison between the observed retention time and those of the standards (37 component FAME mix, Sigma), and the content of each fatty acid was calculate by area normalization method. So, there are some small peaks which occupy the area. But it was not identified by the standards (37 component FAME mix, Sigma), and we had tried to identify these components by GC-MS. There are some chemicals, contained hydroxy, which are not familiar fatty acids.
Minor corrections:
Lines 41-41-“ AA cannot be synthesized by humans and mammals, and it must be provided by exogenous sources, such as food supplement “Rephrase this sentence for example as following: “AA can come from exogenous sources but can also be synthesed from the essential FA, then 18:2 FA provided by diet.
Response: This sentence has been revised, as suggested, and in revised manuscript the sentence has been changed to “AA can come from exogenous sources, and can also be synthesed from the essential fatty acid, linoleic acid (LA; 18:2n-6), which is provided to humans and mammals [3]”
-Line 48. Correct to “…in health….”
Response: This has been corrected in the revised manuscript.
-Line 63-64- Why Comparing with macroalgae Gracilaria Verrucosa? Delete this sentence because micro and macroalgae have quite different biochemical composition.
Response: This sentence has been deleted in the revised manuscript, as suggested.
-Line 121- Growth rate was decreased after 9 days of cultivation not only under Bl-300 but also under L 300.
Response: This has been revised, and in the revised manuscript, the sentence has been changed to “Under the cultivation of L. bisecta SAG2043, the growth rate was decreased after 9 days.”
-line 201-202- need references
Response: The reference has been added in revised manuscript.
- Tables 1-5 :indicate how FA are expressed in the legends. Units are missing. Include also MUFA content.
Response: These have been added in revised manuscript.
Line 181 and others and Fig 2. Indicate what was the light intensity used
Response: The light intensity used (bl-300 for the treatment of culture medium replacement) has been added in revised manuscript.
Line 274” In order to explain such changes, the proportion of AA in the lipid
275 content was determined” did you mean lipid components?
Response: This has been corrected to lipid components in the revised manuscript.
Line 510- GC-FID or GC-MS?
Response: It is GC-FID, and it has been specified in the revised manuscript.
Reviewer 2 Report
Manuscript marinedrugs-757656 "Evaluation and transcriptome analysis of the novel oleaginous microalga Lobosphaera bisecta (Trebouxiophyceae, Chlorophyta) for arachidonic acid production" by Gao et al. does an evaluation of biomass productivity, lipid production fatty acid composition in a novel strain. In addition, they used transcriptome analysis to produce an in-depth look at the biosynthesis of lipids in L. bisecta. Overall the manuscript has merit for publication.
Specific comments:
Line 29 & 104 in the abstract, the authors comment on the "rapid growth" or "growth rate". However, no specific growth rates are presented outside the biomass productivity in g/L. It would strengthen this argument to calculate and give the maximum specific growth rate for each treatment.
Figure 1 In the initial growth period up to about 7-8 days, there is minimal deviation in growth amongst the nitrogen treatments. BG11 is considered to be a nitrogen-rich media. What was the basis for selecting a nitrogen concentration of 3.6mM as low? It would be expected that a more substantial separation in growth would have been observed earlier on in the growth period if the nitrogen was suboptimal.
Tables 1-5 Include the units for the fatty acid measurements.
Line 206 The FAME analysis is represented at % total fatty acids. When summing up the individual fatty acids, the summed values do not equal 100%. Why are the remaining fatty acids in most cases ~5% not reported? Of particular concern is in Table 1 day 3 the values only sum to 80.88%. What comprises the remaining ~19%? Please address this and check on FAME tables.
Line 182 Consider using these treatment labels in addition to those on the graphs to make the text and graphs easier to refer to back and forth.
Line 302 Section 2.1.5 throughout this section as well as others, the authors use the adjective "significant". However, the materials and methods section does not have details on a statistical analysis being performed. Please include this information and where appropriate update figures and tables with relevant information on the statistical thresholds and provide appropriate designations (bold, posthoc, asterisk, etc..) to identify those values that are significant.
Line 466 The authors report a starting inoculation density of OD750 of 0.7. This density is quite high, and self-shading should be a concern. Why weren't the cultures started at a lower density to adequately capture optimal growth and photosynthesis? These higher initial ODs would impact the early growth period results from the different light treatments.
Line 508 Add the concentration of internal standard.
Author Response
Manuscript marinedrugs-757656 "Evaluation and transcriptome analysis of the novel oleaginous microalga Lobosphaera bisecta (Trebouxiophyceae, Chlorophyta) for arachidonic acid production" by Gao et al. does an evaluation of biomass productivity, lipid production fatty acid composition in a novel strain. In addition, they used transcriptome analysis to produce an in-depth look at the biosynthesis of lipids in L. bisecta. Overall the manuscript has merit for publication.
Response: Thanks for your valuable comments on our manuscript. The manuscript has been substantially revised according to the suggestions.
Specific comments:
Line 29 & 104 in the abstract, the authors comment on the "rapid growth" or "growth rate". However, no specific growth rates are presented outside the biomass productivity in g/L. It would strengthen this argument to calculate and give the maximum specific growth rate for each treatment.
Response: I am sorry for our imprecise wording. We revise the rapid growth to high biomass production.
Figure 1 In the initial growth period up to about 7-8 days, there is minimal deviation in growth amongst the nitrogen treatments. BG11 is considered to be a nitrogen-rich media. What was the basis for selecting a nitrogen concentration of 3.6 mM as low? It would be expected that a more substantial separation in growth would have been observed earlier on in the growth period if the nitrogen was suboptimal.
Response: In our preliminary experiment, we measured the nitrogen concentration of day 15 under ul-300, and the nitrogen was depleted under 18 and 3.6 mM treatment in our culture system. Nitrogen concentration of 3.6 mM is low relative to the 18 mM, which is only 20% of initial nitrogen concentration of BG-11 media.
Tables 1-5 Include the units for the fatty acid measurements.
Response: These have been added in revised manuscript.
Line 206 The FAME analysis is represented at % total fatty acids. When summing up the individual fatty acids, the summed values do not equal 100%. Why are the remaining fatty acids in most cases ~5% not reported? Of particular concern is in Table 1 day 3 the values only sum to 80.88%. What comprises the remaining ~19%? Please address this and check on FAME tables.
Response: For the FAME analysis by GC-FID, each fatty acid was identified by comparison between the observed retention time and those of the standards (37 component FAME mix, Sigma), and the content of each fatty acid was calculate by area normalization method. So, there are some small peaks which occupy the area. But it was not identified by the standards (37 component FAME mix, Sigma). We noticed the values of Table 1 day 3 and we had tried to identify these components by GC-MS. There are some chemicals, contained hydroxy, which are not familiar fatty acids.
Line 182 Consider using these treatment labels in addition to those on the graphs to make the text and graphs easier to refer to back and forth.
Response: Thanks for your suggestion. We use the same labels in the text and on the graphs in revised manuscript to make them easier to read.
Line 302 Section 2.1.5 throughout this section as well as others, the authors use the adjective "significant". However, the materials and methods section does not have details on a statistical analysis being performed. Please include this information and where appropriate update figures and tables with relevant information on the statistical thresholds and provide appropriate designations (bold, posthoc, asterisk, etc..) to identify those values that are significant.
Response: Statistical analysis has been added in the revised manuscript.
Line 466 The authors report a starting inoculation density of OD750 of 0.7. This density is quite high, and self-shading should be a concern. Why weren't the cultures started at a lower density to adequately capture optimal growth and photosynthesis? These higher initial ODs would impact the early growth period results from the different light treatments.
Response: As you say, the starting inoculation density of OD750 of 0.7 is quite high compared to many other literatures. In our culture, the OD750 of 0.7 is high under unilateral low light illumination of 100 μmol m-2 s-1, but it is not high under unilateral high light illumination of 300 μmol m-2 s-1 and bilateral high light illumination of 300 μmol m-2 s-1. If the initial microalgal density is too low, the cell will be damaged under high light intensity.
Line 508 Add the concentration of internal standard.
Response: These have been added in revised manuscript.
Reviewer 3 Report
In this manuscript (ID: marinedrugs-757656) the authors examine the most suitable condition to optimize arachidonic acid (AA) 
and lipids production in the coccoid green microalga Lobosphaera bisecta SAG2043. In particular the effect of light intensity and nitrogen concentrations were analyzed. In addition by using a transcriptomic approach they reconstructed the lipid metabolic pathways of L. bisecta for the first time, and demonstrated that up-regulation of keys enzymes was conducive to the accumulation of fatty acids toward AA synthesis. 

The proposed study is interesting and suggest that this oleaginous microalga 
could by a promising candidate to produce AA. This may be an important result for the community. I think that the subject of the paper is well suited for this journal. However the authors’ conclusions are not always supported by experimental evidences and some results should experimentally validate. Same results presented are not robust, I therefore think that the paper need same revision before to bee accepted for the publication.
More specific comments:
- The quality of the figures are not suitable for publication. The resolution of the figures should be improved.
- I suggest to simplify the data reported in the tables 1 to 5, I found the informations reported into the tables to dense. I suggest the authors to
put some information in the supplementary materials - Thanks to the transriptomic analysis the authors reconstructed the lipid metabolic pathways of L. bisecta. I think that, in order to support the conclusions, the authors should experimentally validate the expression levels of some of the genes 
encoding for key enzymes identified (e.g. ACCase) by RT-qPCR.
Author Response
In this manuscript (ID: marinedrugs-757656) the authors examine the most suitable condition to optimize arachidonic acid (AA) and lipids production in the coccoid green microalga Lobosphaera bisecta SAG2043. In particular the effect of light intensity and nitrogen concentrations were analyzed. In addition by using a transcriptomic approach they reconstructed the lipid metabolic pathways of L. bisecta for the first time, and demonstrated that up-regulation of keys enzymes was conducive to the accumulation of fatty acids toward AA synthesis.
The proposed study is interesting and suggest that this oleaginous microalga could by a promising candidate to produce AA. This may be an important result for the community. I think that the subject of the paper is well suited for this journal. However the authors’ conclusions are not always supported by experimental evidences and some results should experimentally validate. Same results presented are not robust, I therefore think that the paper need same revision before to bee accepted for the publication.
Response: We thank the reviewer for the kind words, and constructive comments. These improved the manuscript substantially.
More specific comments:
The quality of the figures are not suitable for publication. The resolution of the figures should be improved.
Response: The figures 1-6 we draw was by Origin 8.5, and the resolution of the figures was 300 dpi. Now, we improved them to 600 dpi in revised manuscript.
I suggest to simplify the data reported in the tables 1 to 5, I found the informations reported into the tables to dense. I suggest the authors to put some information in the supplementary materials.
Response: Thanks for your suggestion. Due to too many data, we draw a figure (Figure 3 in revised manuscript) to replace the table 1-3, and Table 4, 5 was presented as Table S1, S2.
Thanks to the transriptomic analysis the authors reconstructed the lipid metabolic pathways of L. bisecta. I think that, in order to support the conclusions, the authors should experimentally validate the expression levels of some of the genes 
encoding for key enzymes identified (e.g. ACCase) by RT-qPCR.
Response: Thanks for your suggestion. We added the experiment of RT-qPCR to validate the expression levels of some of the genes, mainly the gens encoding desaturases, and these genes are the ones we focus on in this manuscript (Table S4, Figure S1).
Round 2
Reviewer 1 Report
The authors addressed the majority of requested issues, however they did not reply to query that concern the Results of the section 2.1.4.(2.1.4 The proportion of AA within the lipid components of L. bisecta SAG2043) since they did not provide evidence of the identity of the classes of lipids in fractions or if they were able to seprated tehm effectively. They could have a mix of different classes in the same fraction .
. Thus, if no additional work will be done, this section 2.1.4 must be remove as well as discussion and conclusion addressed based on this results.
Author Response
Response to Reviewer 1 Comments
The authors addressed the majority of requested issues, however they did not reply to query that concern the Results of the section 2.1.4.(2.1.4 The proportion of AA within the lipid components of L. bisecta SAG2043) since they did not provide evidence of the identity of the classes of lipids in fractions or if they were able to seprated tehm effectively. They could have a mix of different classes in the same fraction .
Thus, if no additional work will be done, this section 2.1.4 must be remove as well as discussion and conclusion addressed based on this results.
Response: Thanks for your suggestion. I am sorry that I did not make my reply clear to your query that concern the lipid classes. The aim of lipid classes analysis was to know that why the AA content increased along with the total lipid content increasing. We speculated that AA accumulated in storage lipid not in membrane lipid. So, we just do the preliminary fractionation of lipid extracts to separate the total lipid to neutral lipid (storage lipid), glycolipids and phospholipids (major constituents of thylakoids, plasma membranes, and endoplasmic membrane systems, which perform a structural role). The method of using solid-phase silica gel column to separate lipid classed was mature. We used thin layer chromatography analysis to identify neutral lipid (Figure). Neutral lipids was mainly constitute of TAG, with some content of DAG, not with polar lipid. If you insist that we should remove the section 2.1.4, we will delete this part.
Figure (Please see the file attached). Thin layer chromatography analysis of total lipids and neutral lipids (: TAG, triglyceride; DAG, diacylglycerol; FFA, free fatty acids; glycolipids, GL; phospholipids, PL).

Reviewer 3 Report
The authors have satisfactorily responded to all my questions and made the necessary changes to the manuscript. However, others questions occurred to me in reading the revised manuscript, which would be incorporated to finalize the manuscript.
Specifically: as requested the authors performed the experiment of RT-qPCR to validate the expression levels of some of the genes, mainly the genes encoding desaturases as reported in Figure S1. However in the legend of the figure the authors should specify the full name of the genes analyzed (e.g is not clear to me if KAS refer to KASI or II) and as well the full description of the samples they tested.
In addition since they performed the analysis, for each sample, in triplicate they should add the standard deviation.
In the text, Line 392 ,the authors report: “the expression of 7 genes were up-regulated and 3 genes were down-regulated under the condition of nitrogen deficiency (Table S4, Figure S1)”, however in the Figure S1
I see the result reported only for 5 genes. Please correct or explain better.
Author Response
Response to Reviewer 3 Comments
The The authors have satisfactorily responded to all my questions and made the necessary changes to the manuscript. However, others questions occurred to me in reading the revised manuscript, which would be incorporated to finalize the manuscript.
Specifically: as requested the authors performed the experiment of RT-qPCR to validate the expression levels of some of the genes, mainly the genes encoding desaturases as reported in Figure S1. However in the legend of the figure the authors should specify the full name of the genes analyzed (e.g is not clear to me if KAS refer to KASI or II) and as well the full description of the samples they tested.
Response: I am sorry for our carelessness. It has been added in revised manuscript. Figure S1: The gene expression validated by Quantitative realtime-PCR; KASII, 3-ketoacyl-ACP-synthase II; thioe, thioesterase; 9 des, Δ9-desaturase; 6 des, Δ6-desaturase; 5 des, Δ5-desaturase; d3-NR, cultivation time of day 3 under 18 mM nitrogen concentration; d3-NF, cultivation time of day 3 under 0 mM nitrogen concentration; d6-NR, cultivation time of day 6 under 18 mM nitrogen concentration; d6-NF, cultivation time of day 6 under 0 mM nitrogen concentration.
In addition since they performed the analysis, for each sample, in triplicate they should add the standard deviation.
Response: It has been presented in revised Figure S1.
In the text, Line 392 ,the authors report: “the expression of 7 genes were up-regulated and 3 genes were down-regulated under the condition of nitrogen deficiency (Table S4, Figure S1)”, however in the Figure S1, I see the result reported only for 5 genes. Please correct or explain better.
Response: “the expression of 7 genes were up-regulated” In this sentence, “the gene” was unigene. There were 5 unigenes identified as Δ9-desaturase. When we do the qPCR, we only choose one unigene of Δ9-desaturase which expression level was high to make sure the experiment is successful.

Round 3
Reviewer 1 Report
The authors answered the question asked by the reviwer and showed a method (TLC) that allows to confirm that there was separation of the classes and pinpointing it identity.
The authors must acquire a new TLC image in which they show, in two new lines of the TLC, the fractions of PL and GL obtained by SPE.
This new image must be included in the manuscript or added as supplementary information
Author Response
Response to Reviewer 1 Comments
The authors answered the question asked by the reviwer and showed a method (TLC) that allows to confirm that there was separation of the classes and pinpointing it identity.
The authors must acquire a new TLC image in which they show, in two new lines of the TLC, the fractions of PL and GL obtained by SPE.
This new image must be included in the manuscript or added as supplementary information.
Response: Thanks for your suggestion. New figure (Figure S1) is added in revised manuscript. “Each lipid fraction was confirmed by thin-layer chromatography (TLC) on silica gel. The developing solvent for NLs was hexane: diethyl ether: acetic acid (80: 20: 1). The mobile phase for GLs and PLs was chloroform: methanol: H2O (25: 10: 1).” is added in Methods section.
Figure S1: Separation of lipid fraction by thin-layer chromatography (TLC). a, separation of neutral lipids (NLs); b, separation of glycolipids (GLs) and phospholipids (PLs); TAG, triacylglycerol; DAG, diacylglycerol; FFA, free fatty acid.